# A Study on the Visual Communication and Development of Green Design: A Cross-Database Prediction Analysis from 1972 to 2022

**Yixuan Du** [1] and **Hailan Ma** [2],*

---

1   Academy of Arts & Design, Tsinghua University, Beijing 100190, China; duyixuan@mail.tsinghua.edu.cn
2   Faculty of Foreign Languages, Guangdong Ocean University, Zhanjiang 524088, China
*   Correspondence: noname1593@stu.gdou.edu.cn

**Abstract:** This paper resulted from 1775 pieces of literature from the WoS database and 1923 from the CNKI database. The research framework, development process, internal relations, and key hot topics of green design were explored through the tool of CiteSpace metrology. Four main research results were presented: (1) Green design began to grow explosively around 2015. The design revolved around energy saving, material selection, and other aspects, with a high rate of cooperation, high reference, a large number of publications, and other upsurge phenomena in each branch of the formation. (2) There is the highest volume and popularity of research in the CNKI database. At the same time, in WoS, the United States still has the most disciplinary influence and academic exchange freedom. (3) WoS focuses on solving practical problems of branch disciplines, mainly chemical engineering, experiments, and case analyses. The hot topics in CNKI tend to be design subjects, mainly design technology, management, and theory. (4) There are few basic types of research on the WoS database; CNKI pays more attention to design theoretical research. The two databases form complementary solutions to ensure the future development of green design. The results indicate that green design should be envisioned as an eco-friendly approach, emphasizing optimizing human and management practices, innovative design principles, sustainable processes, and consideration of sociocultural impacts.

**Keywords:** green design; sustainable development; green idea; cluster analysis; prediction analysis

## 1. Introduction

The motivation for this study is to elucidate the developmental trajectory of green design and pinpoint its research hotspots. Previous studies have examined the impact of green design on the field from various professional perspectives. With green design as the central focus, scholars in architectural design [1], manufacturing [2], life cycles [3], and operational models [4] emphasize the significance of green design for promoting sustainable development and enhancing human well-being. Utilizing a quantitative econometric approach, this paper aims to reveal the framing issues of the discipline. This methodology also enables a comprehensive understanding of the field's research direction, opportunities, and challenges from a macro perspective.

The aim of the present study is to provide an objective analysis of the future prospects of green design. It employs a comprehensive, cross-database methodology to augment the breadth, heterogeneity, and scope of literary information pertaining to eco-friendly design. Because of the different databases and languages used, it is easy to overlook green design research in China. Therefore, to achieve a comprehensive and substantial review of green design, this study not only gathers data from the Web of Science database but also incorporates CSSCI Chinese literature from the China National Knowledge Infrastructure (CNKI) to visualize and analyze key factors such as abstracts, keywords, author relationships, and national institutions in the relevant literature.

By employing a cross-database approach and utilizing CiteSpace software to examine the knowledge structure and developmental lineage of the literature, we can clearly identify the prevalent research trends and future directions in the field of green design. Based on the quantitative results generated by CiteSpace software using an econometric approach, this visual analysis illustrates the development of green design history and current research hotspots and predicts future development directions. Furthermore, it serves as a valuable supplement to green design research in China. Consequently, well-founded insights into the future direction of green design, both in China and globally, are offered by facilitating the formulation of plans for future study.

## 2. Literature Review

The study of green design, championed by industrial designers and artists, emerged in the 1990s as a reflective design philosophy addressing human, societal, and environmental concerns [5]. This concept has influenced various aspects of art and design, serving as a green principle in fields such as design, engineering, chemistry, and aesthetics. Green design differs from environmental and sustainable design as it seeks innovative solutions or alternatives to traditional approaches—design solutions that address the environmental crisis [6]. In contrast, the term "eco-design" is not as widely recognized, and a cost-effective design strategy focusing on reducing pollution from industrial production has been proposed [7]. Green design prioritizes harmony with nature and has emerged as a significant research concept [6]. Designers committed to sustainable development have adopted green design to innovate traditional design concepts, material usage, and technical methods. The concept emphasizes the core design objectives of reducing environmental impact and fostering protection [8]. In essence, green design is one of the most crucial initiatives for carbon-neutral development in the present day [9].

However, the rapid advancement of green design also presents various problems and challenges. Numerous issues arise from green design concepts [10], such as the over-packaging of green ideas, irrational use of materials, greenwashing concerns, green fraud, and the limitations of green design in long-term considerations. Ultimately, green design must balance market, environmental, and design elements to find effective solutions. Moreover, green design is primarily implemented in production facilities and businesses [11]. While it is a research approach based on product processing and practice, employing design methodologies and disciplinary perspectives, the fundamental theoretical research requires further exploration to solidify the underlying principles [12].

To achieve the objectives of this study, it is first necessary to overview the development process and main concepts of green design. It is essential to consider whether this results from considering market demand and the natural environment. Then, to achieve the results by making the appropriate design sacrifices or changes in design direction [13]. If the market demand (economic efficiency) is overly considered, it may lead to greenwashing problems such as over-promotion of environmental protection [14]. On the other hand, over-considering the impact of materials on the natural environment can lead to a waste of resources due to over-experimentation, which in itself may be more costly than not replacing materials in the natural environment [15]. The amount of control between the two is green design, which means that design is the critical research object that can determine whether green design can implement sustainability. Therefore, clarifying the current status and characteristics of green design research can facilitate the advancement of green design and thus provide a more rational approach to the future trend of sustainable development.

The core concept of the "design" discipline is "human-centeredness" [16]. Countries worldwide adapt their design approaches to their specific conditions and circumstances, aiming to develop the most suitable local design concepts that yield reasonable results or products [17]. Green design focuses on reducing environmental costs and minimizing environmental impact through design, using available resources as a foundation. While the design may seem like a small part of the production, the comprehensive implementation of green design necessitates making it a top priority. To develop a service policy for green

design, the environmental impact of production must be considered, from raw materials to production processes, encompassing a range of issues such as material selection and experimental processing [18]. Consequently, the significance of green design research lies in promoting sustainable development from the source and forming an integral part of the entire eco-production chain in the future [19].

Green design varies globally due to various social needs, historical backgrounds, current issues, and primary conditions. As a result, the applications, approaches, concepts, and concerns related to green design also differ. Designers in developed Western countries were among the first to reject the notions of "luxury", "extravagance", or "sophistication", becoming aware of the damage industrial products inflicted on the natural environment. This awareness led to the development of green design's ethical principles [20]. A key advantage of green design is its cost-effective ability to address problems such as environmental pollution and waste. It plays a particularly crucial role in achieving sustainable development goals for developing countries by emulating viable solutions and efficiently enhancing ecological conditions.

In recent years, China has placed significant emphasis on the impact of production on environmental pollution in its development efforts. In 2017, for the first time, the Chinese government identified green development as one of the core objectives of national development [21]. Furthermore, in 2021, the government announced its intention to achieve peak carbon dioxide emissions by 2030 and carbon neutrality by 2060 [22]. These commitments have positioned China as a favorable research base and established a healthy development trend in the field of green design in recent years.

### 3. Methods and Data

CiteSpace software helps visualize the abstract citation network, understand the knowledge structure in more detail, and explore its development patterns and trends [23]. Based on methods such as co-citation path Finder, CiteSpace software performs econometric projections of key hotspots and knowledge inflection points to explain and anticipate the phenomena that occur in the field [24]. Figure 1 shows that the study was divided into data collection, analysis, and result and conclusion. First, valid literature from two databases has been integrated, and variable elements, such as title, author, country, keyword, abstract, and affiliation, are refined. After that, a visual clustering graph is generated by analyzing the vertices and determining the cluster community in the data [25] Next, results were generated based on temporal variables, including keyword and knowledge graphs. Finally, the results combined with literature reviews for the development process of the subject area, clustering relationships, authors, and affiliations. A comprehensive picture of green design research was presented and complemented with a presentation of green design in China.

CiteSpace software was adopted in this study to do the research literature on green design at home and abroad. The Web of Science database (WoS) was used as a research sample for global literature data. Journal search types were selected as "Social Sciences Citation Index (SSCI), Sciences Citation Index (SCI), and Humanities Citation Index (A&HCI)" to ensure the relevance and scientific quality of the literature.

The title was set as "Green design" and the sample period was from January 1972 to December 2022, but excluded art exhibition reviews, news, and secretary chapters. G-index = 25, the pruning strategy was set to the relationship between Pathfinder and Pruning Sliced Network, the co-citation rate (ccv) and the lowest citation or occurrence frequency (c), and co-citation frequency (cc) in this time slice was $ccv(i,j) = \frac{cc(i,j)}{\sqrt{c(i)*c(j)}}$, and the time slice was 1 year. Chinese data were selected from CNKI database; considering the seriousness of the study, only CSSCI and Chinese core periodicals (PKU) were selected as journal types and were set as "green design or green art design", with 1923 samples from January 1993 to December 2022. If G-index = 5, the branching strategy was set to Pathfinder and Pruning Sliced Network, with time slices of 1 year.

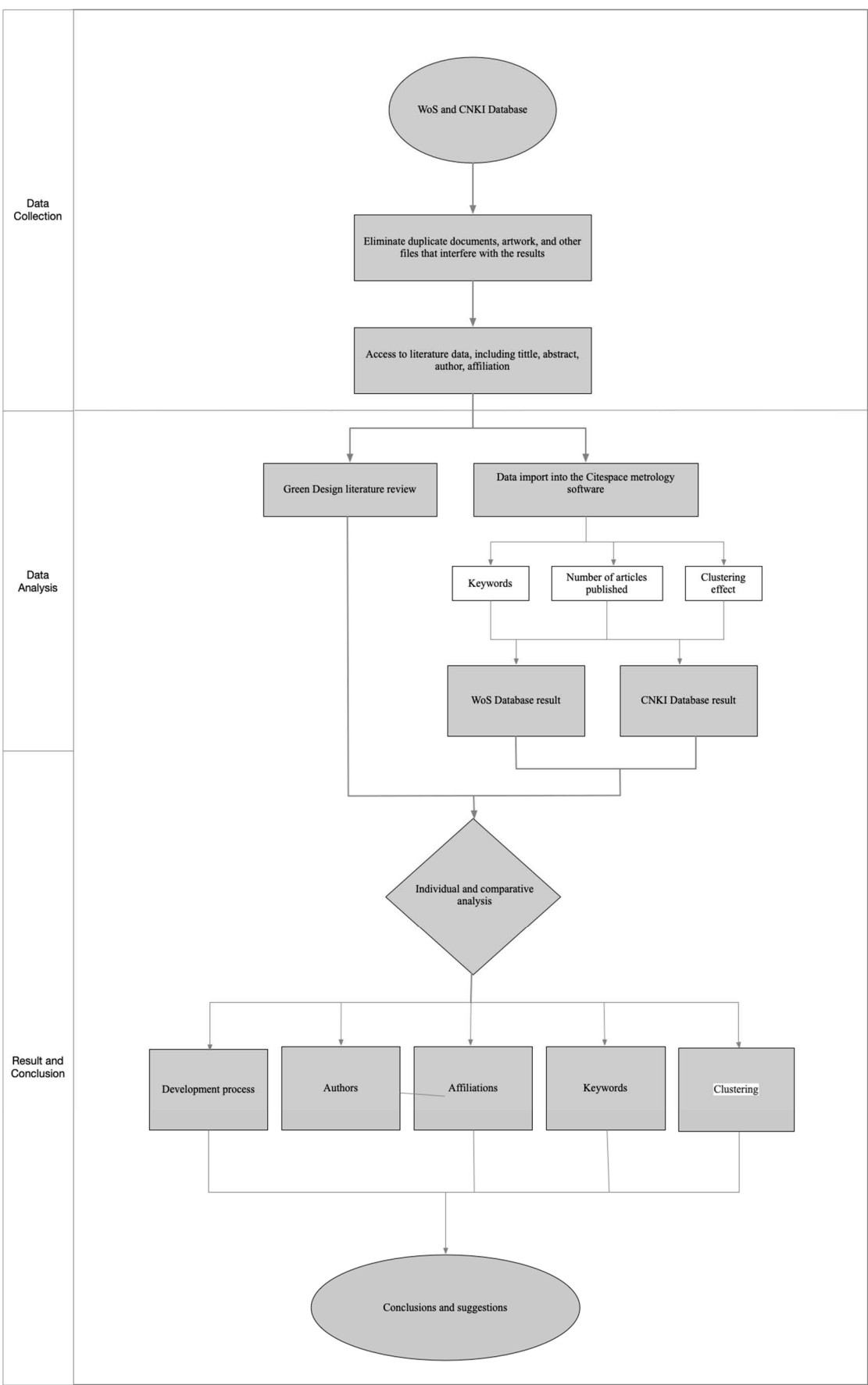

**Figure 1.** Overview flowchart.

## 4. Number of Literature Publications

CNKI and WoS databases show that from the appearance of the first published paper in 1972 to the present (2022), the total number of green design publications was 3698, of which 1923 were in the CNKI database, and 1775 were in the WoS database. The totality of publications is a visual way to understand the overall development process of green design, which is of practical significance to the discipline's history and dynamics.

The first academic paper on green design was published in 1972, according to Figure 2. By 2022, green design will have been studied and investigated for 50 years, with 1775 published articles. In terms of its development process, three phases were formed, with 2002 and 2014 as its demarcation points. The first phase was from 1972 to 2002, with a low annual publication volume of about five articles. The second phase was from 2003 to 2014, with an increase in the number of yearly articles compared to the previous period. The average annual number was about 35 articles, with a growth period belonging to the theme of green design. Finally, the third stage, from 2015 to 2022, is the explosive period of the theme, with an average annual volume of 154 articles, among which the annual publication reached the highest of 253 articles in 2022. Based on the above three stages, it can be seen that green design is developing on a steady upward trend in international development, and the attention grows with the annual growth. Therefore, the theme has become a hot topic in recent years.

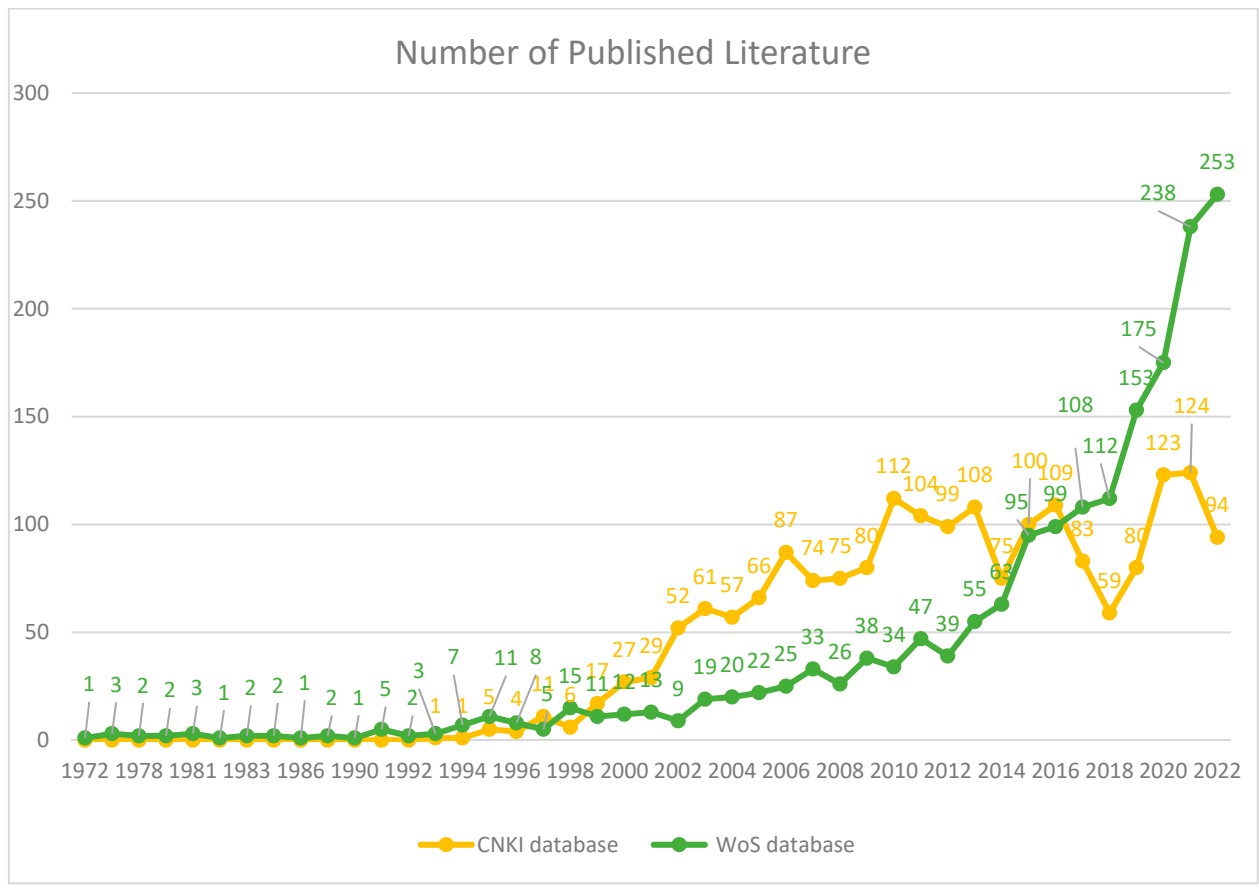

**Figure 2.** The trend of green design publications in CNKI and WoS database.

The first academic literature on green design in China, as shown in Figure 2, was published in 1993, with 1923 publications over 29 years (1993–2022), divided into three stages, with 2002 and 2010 as the cut-off points. The first phase, from 1993 to 2001, was the budding period of this research topic. The number of articles published was relatively low, at about 10 per year. Over time, the number of articles grew slowly to about 20 per year. Scholars paid little attention to it, and there is no hot topic in the discipline. In the

second period, from 2002 to 2010, the topic experienced explosive growth, and the number of articles exhibited an overall growth view, from about 60 to 100 articles per year. During this period, scholars paid attention to the green design research topics and formed a high research enthusiasm. The third period was from 2011 to 2022, with a relatively stable number of articles, around 110 per year. During the fluctuation from 2017 to 2019, the number of articles declined significantly. However, in 2020, it rose again and reached the highest annual number of 124 articles in 2021. At this stage, green design has become a stable research trend in China.

Compared with the number of domestic and foreign publications, the average number of international publications in the WoS database in the past 50 years is about 36, and that of China in the past 29 years is about 83. Regarding the overall trend, globally, green-related topics are hotter earlier in China than in the international setting, and WoS's hot burst is higher than China's. Regionally, China has become one of the leading countries for research on green design.

## 5. Regional and Author Distribution Analysis

Co-citation, different from citation, is when two or more literature cites one literature at the same time. It is a tool to directly confirm the internal relationships between disciplines and follow the field's development through clustering [26]. Therefore, the co-citation approach can effectively present the relationship between institutions (Figure 3) and the number of authors co-cited (Table 1).

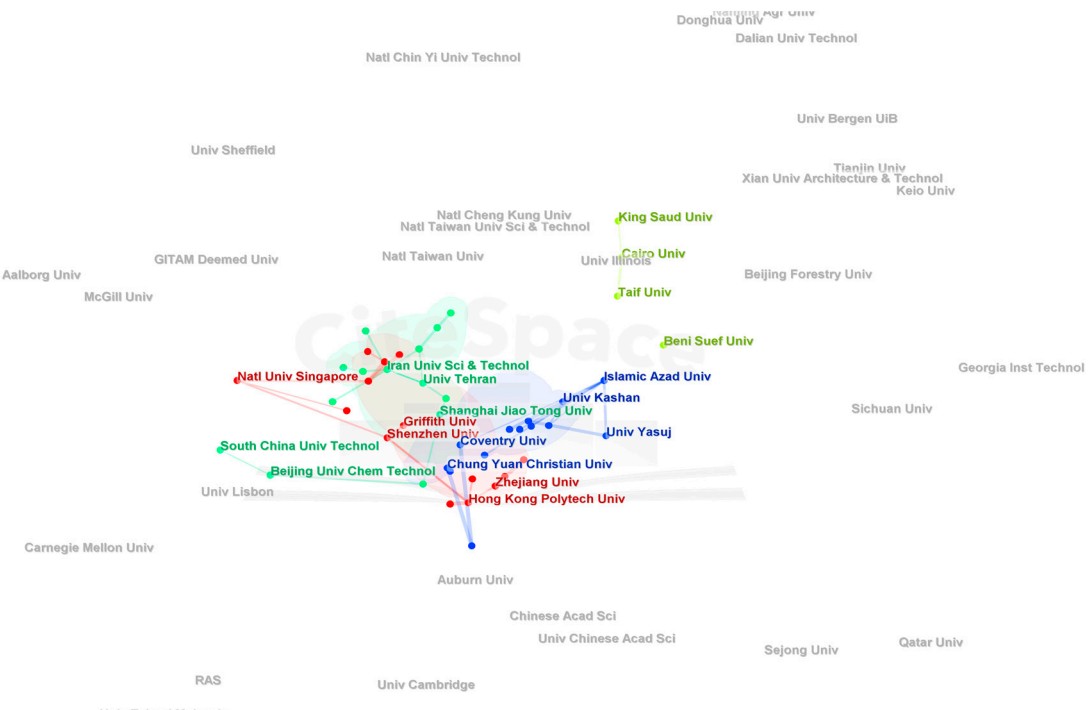

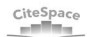

**Figure 3.** Institution relationship network.

**Table 1.** Authors' total co-citations.

| Authors | Count | Year |
| --- | --- | --- |
| Zhang, Liqun | 5 | 2019 |
| Green, JE | 4 | 2002 |
| Yeang, Ken | 4 | 2007 |
| Kowtharapu, Leela Prasad | 4 | 2022 |
| Sandoval, Christian A | 4 | 2022 |
| Katari, Naresh Kumar | 4 | 2022 |
| Anastas, Paul T | 3 | 2009 |
| Anastas, PT | 3 | 2000 |
| Rekulapally, Vijay Kumar | 3 | 2022 |
| Ghaedi, M | 3 | 2016 |

On the other hand, as shown in Figure 3, academic institutions in some countries or regions have formed a cooperative relationship network with joint research behaviors and mutual citation, which is detailed as follows: Firstly, there is Zhejiang Univ (8 times), Hong Kong Polytech Univ (15 times), Shenzhen Univ (10 times), Griffith Univ (3 times), cooperated with Natl Univ Singapore (7 times) and other institutions in the red relationship chain, and they formed cooperative relationships with a total of 43 co-citations. For example, Griffith Univ and Shenzhen Univ collaborate on sustainability and regenerative design for green buildings [27]; Hong Kong Polytech Univ and Zhejiang Univ collaborate on life cycle design in reinforced concrete structures [28]; Natl Univ Singapore, Shenzhen Univ, and Hong Kong Polytech Univ are collaborating on green building benefit mechanisms and partnerships [29].

Second, the blue relationship chain includes Islamic Azad Univ (9 times), Univ Kashan (2 times), Univ Yasuj (4 times), Coventry Univ (5 times), and Chung Yuan Christian Univ (5 times) with a total of 25 co-citations. For example, Univ Yasuj and Islamic Azad Univ collaborate on optimizing wastewater for green adsorption production and processing processes [30]; Coventry Univ and Islamic Azad Univ collaborate on research on reverse supply chain network issues in green supply chains [31].

Third, the green relationship chain includes South China Univ Technol (2 times), Beijing Univ Chem Technol (7 times), Iran Univ Sci & Technol (7 times), Univ Tehran (10 times), Shanghai Jiao Tong Univ (3 times) and other institutions with 29 co-citations. For example, South China Univ Technol and Beijing Univ Chem Technol cooperate in green research recycling, green cross-linking recycling, and an actual recycling loop [32]; Iran Univ Sci&Technol and Univ Tehran have often cooperated in the green supply chain many times, including bi-objective green supply chain network design [33], a green supply chain network design model [34], and a fuzzy pricing model's green competitive [35].

Fourth, the light green relationship chain includes King Saud Univ (5 times), Cairo Univ (5 times), Taif Univ (10 times), and Beni Suef Univ (5 times), with a total of 25 co-citations. For example, King Saud Univ and Cairo Univ collaborate on Green Technologies applied in Jyotishmati [36]; Cairo Univ, Beni Suef Univ, and Cairo Univ collaborate on green and efficient extraction methods from waste materials [37,38].

From the perspective of research institutions [38], green design has a positive partnership. The research institutions are mainly universities, with many research institutions from China and other Asian regions. It indicates that the communication on this topic is biased toward the Asian region. These exchanges help academics influence each other, promote each other, and direct the future development of green design. However, some shortcomings of previous research have been shown in Table 1; the posting authors need to be more scattered to form a clustering connection, and the centrality is 0. It indicates that researchers are mainly individual researchers and do not form a project research team for green design.

## 6. Green Design of WoS Database

### 6.1. WoS Database Keywords

A keyword analysis is a reliable way to obtain the main content of the literature. One-third of readers and researchers can find the correct text using keywords alone [39]. In addition, keywords can be used to understand the main idea, meaning, direction, and other core information of the literature. Moreover, green design requires multidisciplinary involvement, the integration of different technical approaches, and the combination of diverse fields to achieve comprehensive development in different fields. Cluster analysis is a famous software for multidisciplinary participation in diversified analysis in recent years, which can develop and predict topics [40]. After sorting out the classification intensity, word frequency, citations, and clustering of the keywords in the literature samples, a reliable keyword map and cluster map are finally formed.

In Figure 4, the top seven keywords' strengths under the green design theme are presented in order of green design, genetic algorithm, methylene blue, ecosystem service, ion, strategy, and decision. From the strengths and types of keywords, green design mainly acts in the chemical and environmental design directions, such as genetic algorithm, methylene blue, and ion belonging to the chemical subject; the research themes focus on making environmentally friendly products using appropriate materials and methods. The focus is technology, resources, efficiency, management, and macro-environment. Green design keywords have the highest intensity and most extended duration properties. After that, more hot topics have formed in recent years, mainly focusing on the complex key issues of specific implementation methods of green design. Green design has been paid attention to by scholars and scientists in many aspects in recent years, forming a high hot trend of multiple gathering points and aspects.

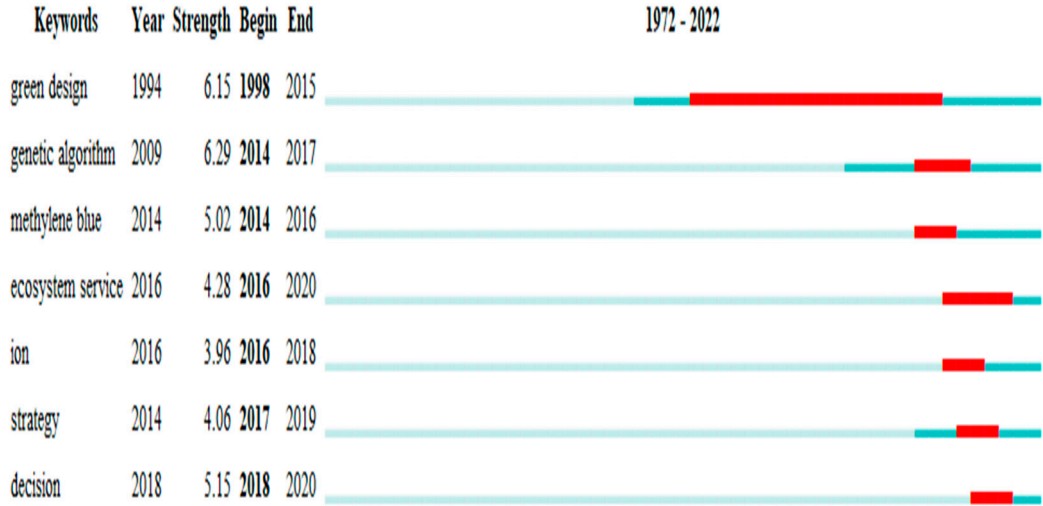

**Figure 4.** WoS database green design keywords.

### 6.2. Green Design Clustering for WoS Database

The clusters' size, number of papers, distance, density, and centrality are all critical in determining the clustering relationships. Table 2 indicates the clustering data where the Silhouette values are all within a reasonable range. Figure 5 shows ten groups of hot topics formed by green design. Each cluster group consists of related classifications, which can refine the analysis of the development trend of green design. The specific cluster group sets in order of cluster size are green supply chain (size 103), green chemistry (size 96), malachite green (size 68), green infrastructure (size 57), green design (size 41),

protein engineering (size 39), enhanced analytical quality (size 34), energy consumption (size 33), life cycle assessment (size 26), and olive leaves (size 25). The figure shows the high clustering density and group sets' position near the center point. This demonstrates a sense of well-organized collaboration and a chain-like research status for green design research. Among them, management disciplines have the most significant research base. Chemical engineering forms the widest variety of clusters. The design disciplines are closely related to the urban environment.

**Table 2.** Ten clustering groups analysis.

| Order | Cluster ID | Size | Silhouette | Mean (Year) |
|-------|-----------|------|------------|-------------|
| 1 | green supply chain | 103 | 0.673 | 2014 |
| 2 | green chemistry | 96 | 0.79 | 2012 |
| 3 | malachite green | 68 | 0.86 | 2012 |
| 4 | green infrastructure | 57 | 0.795 | 2018 |
| 5 | green design | 41 | 0.974 | 2003 |
| 6 | protein engineering | 39 | 0.873 | 2004 |
| 7 | enhanced analytical quality | 34 | 0.842 | 2011 |
| 8 | energy consumption | 33 | 0.901 | 2010 |
| 9 | life cycle assessment | 26 | 0.957 | 2006 |
| 10 | olive leaves | 25 | 0.929 | 2014 |

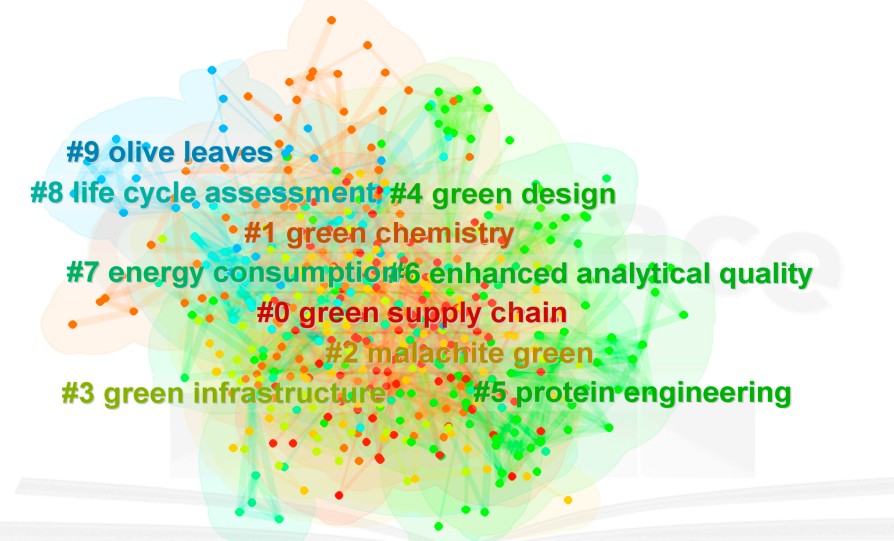
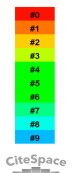

**Figure 5.** Cluster diagram of international green design.

Figure 5 is about ten groups of hot topics formed by green design. Each cluster group consists of related categories, which can refine the analysis of the development trend of green design. The distinct colors represent various clusters, while the density relationships among the points signify the magnitude of their connections. The cluster groups are as follows in the same order as in Table 2.

There are 3 directions in these 10 clusters, namely design direction, chemical engineering industry, and energy management. Firstly, for design direction, environment, product, ecology, and architecture are involved. For example, hospitals used ecological

plant environments to improve quality [41], developed the Design for Environment (DFE) tool to improve the recycling mechanism of waste products [26], energy-saving building materials were also referred to [42], and so on.

Secondly, in terms of the chemical engineering industry, chemistry lab, life cycle assessment, carbon dioxide, phenolic compounds, and an ionic liquid are keywords. For example, proposed a rapid life cycle assessment in the cases of buildings, electromechanical products, and regulators [43]; proposed the BioTriz multi-contradiction resolution method [44]; proposed a new assessment recycling mechanism to achieve life cycle assessment [45]; proposed a quality-by-design (QbD) method from molecular base properties [46]; proposed plastic chemical degradation and other methods to achieve green chemistry [47,48], from low-carbon product design [49], and using low-carbon engineering construction methods and other directions to achieve a low-carbon sustainable path [50].

Thirdly, energy management consists of climate change, supply chain management, risk management, multi-objective optimization, management plan, and other keywords. For example, using a Green Supply Chain Management (GSCM) model in sustainable management [38], developed Green Supply Chain design methods and other implementations in green management [51]. From the three aspects, the key hotspots of green design are the implementation and the specific measures in the generation process. Clustering topics are depicted as terminological names or specific micro cases.

Green design as a subject has not developed prominent research theories and research frameworks. Bridging the framework of green design would bring a more practical meaning to green design. The hot topics have significant disciplinary gaps due to the diverse elements of the topic, but the research has practical guidance for the discipline.

## 7. Green Design in China

### 7.1. China Green Design Keywords

As for green design in China, the methods adopted are similar to the study of international green development topics: keywords, clustering graphs, and knowledge graphs. The top ten intensity keywords under the green design theme are illustrated in Figure 6: green product, environment, concurrent engineering, ecological design, agricultural engineering, energy conservation, green architecture, green chemistry, green ideas, and package design. From the keyword intensity and type perspective, green design in China mainly focuses on practical design to reduce environmental pollution, aiming at reducing environmental damage caused by man-made production behavior. Although green design belongs to design science and is a conceptual design with green as its core, around this core, the concept of green is widely distributed in livelihood industries and production fields, especially in green products and green buildings, and has reached the highest research intensity. After the development of green design to a certain level, its design ideas and theoretical methods have become a hot topic under the green theme in recent years. This means that green design shows the trend of developing disciplinary theories.

## Top 10 Keywords with the Strongest Citation Bursts

| Keywords | Year | Strength | Begin | End | 1993 - 2022 |
|---|---|---|---|---|---|
| Green Product | 1995 | 12.4 | 1995 | 2004 | |
| Environment | 2000 | 6.11 | 2000 | 2005 | |
| Concurrent Engineering | 2003 | 4.24 | 2003 | 2007 | |
| Ecological Design | 2005 | 4.6 | 2005 | 2011 | |
| Agricultural Engineering | 2006 | 4.17 | 2006 | 2007 | |
| Energy Conservation | 2010 | 5.79 | 2010 | 2017 | |
| Green Architecture | 2003 | 21.47 | 2011 | 2017 | |
| Green Chemistry | 2009 | 4 | 2018 | 2022 | |
| Green Ideas | 2012 | 7.04 | 2019 | 2022 | |
| Package Design | 2002 | 3.93 | 2020 | 2022 | |

**Figure 6.** Green Design Keywords in CNKI database.

*7.2. Green Design Clustering in CNKI Database*

According to Figure 7, nine thematic clusters are shown as green design, green manufacturing, green building, green products, green, packaging design, green concept, product design, and overview. The center point of the figure is green design. Unlike the WoS database, the CNKI database is centered on green design and spreads to other disciplines. These nine categories are the nine hot research directions of green topics. The clusters with higher density are shown in the figure to indicate the close connection between industry, manufacturing, and design, which is a concrete representation of the implementation of green design in the industry to reduce the environmental impact. The higher values of outliers in the overview clusters indicate that green design research does not form a scaleable research relationship. Examples of specific cases are Xixi National Wetland Park in Hangzhou, using ecological green design to achieve rational land planning and protection of endangered flora and fauna [52]. The Chinese government enhanced the implementation of green buildings between 2011 and 2015, including eco-city and green building towns [53]. China's steel industry uses energy-saving design, clean production programs, and other methods to achieve green manufacturing [54]. From the psychological surveys on consumer responsibility [55], Chinese traditional culture promotion [56], government support [57], and popular values [58], the green design feasibility is confirmed. In addition, it is proven to be practical from the perspective of artistic design, such as paper packaging for long-life noodles [59] and minimalist design [60].

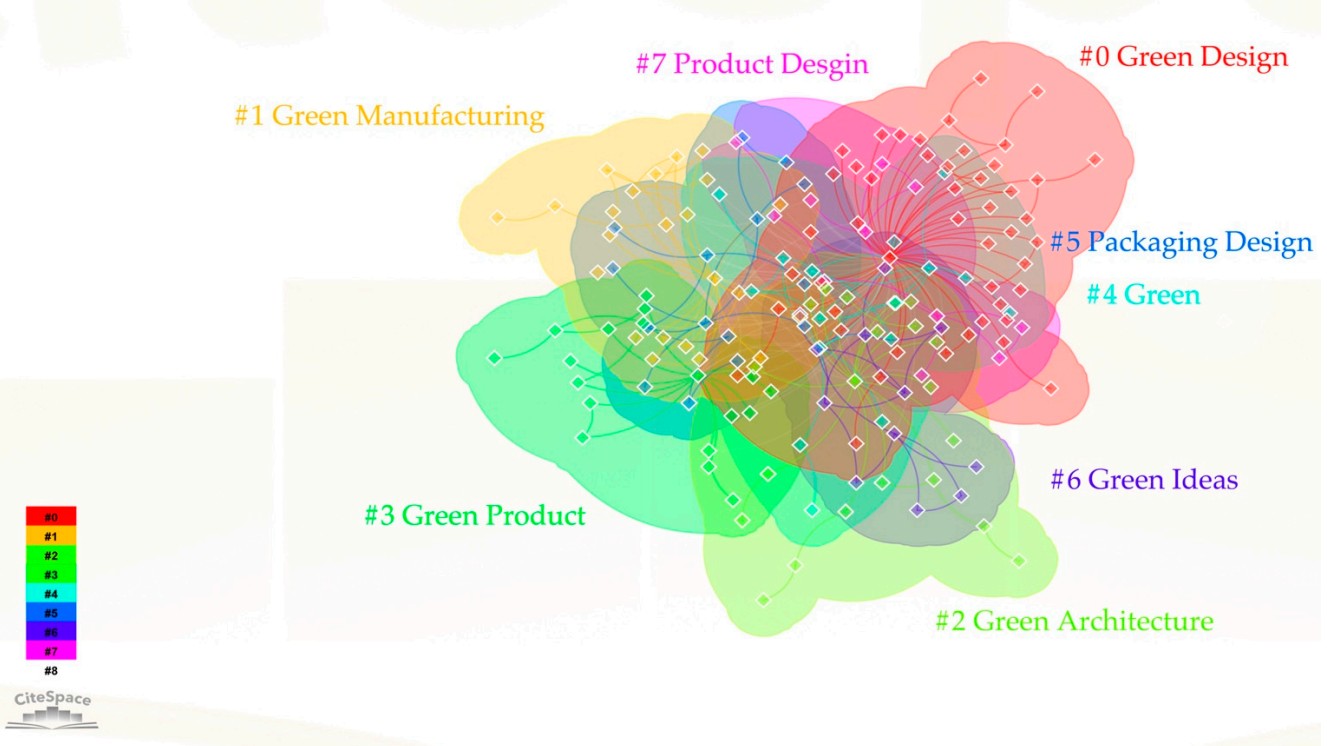

**Figure 7.** Graph Clustering of green design in CNKI database.

Furthermore, the discussion of green design disciplines consisting of overviews and theoretical studies constitutes a significant theme. For example, the use of green concepts to stimulate green development in mountainous and rural areas [61], innovation of green concept aspects through carbon emission analysis [62], complementing the green concept through low carbon tourism [63], and other aspects. According to Figure 7, it can be found that the hotspots of green design in China are mainly in the use of design solutions combined with scientific approaches for concrete example analysis. Unlike the WoS database, green design theory in China has started to form hot topics, which means that

Chinese scholars have begun to focus on discussing the underlying theoretical concepts of green design.

## 8. Analysis of Results

### 8.1. WoS Database Green Design Hotspot Analysis

The knowledge graph is a method that combines large-scale and diversified data to provide reliable research direction for future research topics [64]. For example, in Figure 8, the data sample is the same as the literature data for the cluster diagram. The relationship between the timeline and the clusters is added, and the trend can be analyzed in chronological order.

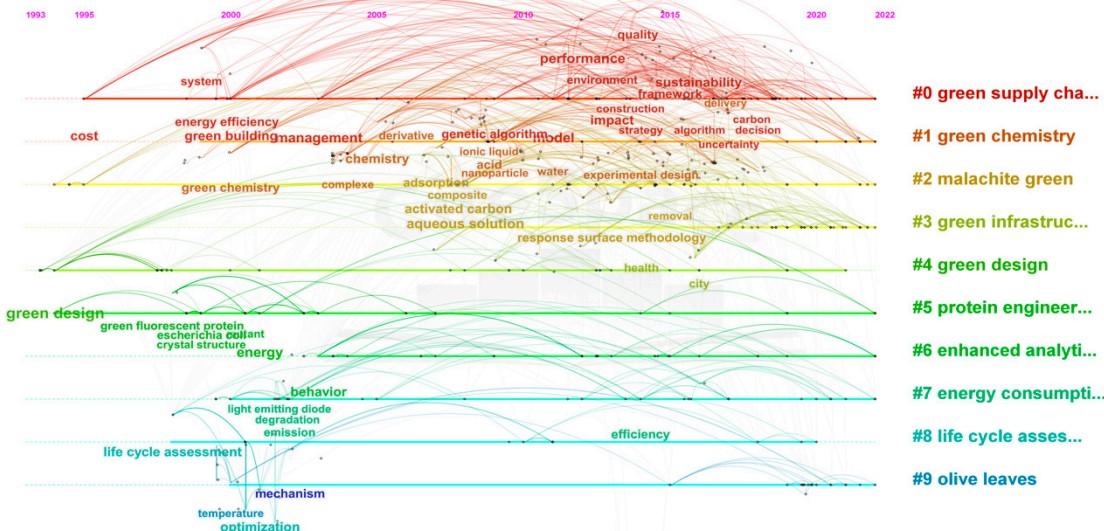

**Figure 8.** WoS database knowledge graph.

Green design is a prominent research topic in the WoS database, with critical issues such as green design and life cycle assessment being the starting points for research. The themes of the green supply chain, green chemistry, and malachite green exhibit high density and proximity, with few outliers, indicating a close relationship between these research clusters. The remaining clusters are relatively independent, with a few high outliers. The overall trend of green design in the WoS database is clear, with ten clusters forming stable hot topics since 1995, while two clusters are no longer hot topics.

Green design encompasses various research approaches, including chemical experiments, case studies, theoretical concepts, and interdisciplinary research. Each sub-theme area has related hot topics, reflecting various disciplines' particular sustainable development research. While the green design may sound like a design concept, the hot topics demonstrate that it encompasses a variety of research fields. Stable hot spots and critical special studies exist under each theme, contributing to the overall advancement of sustainable design.

### 8.2. Green Design Hotspot Analysis in CNKI Database

The sample of data analyzed is the same as the literature data of the cluster graph in the CNKI database. Centroids are green design's most crucial starting point. Next are research data points on green products, environmental protection, and product design. Green design clustering has the highest density and proximity. It indicates that the research primarily conducts with green design as the topic. As shown in Figure 9, the overall trend of green design in China is simple. It consists of three groups of stable research topics that have continued until the present day in terms of hotness: green design, green manufacturing, and green composition.

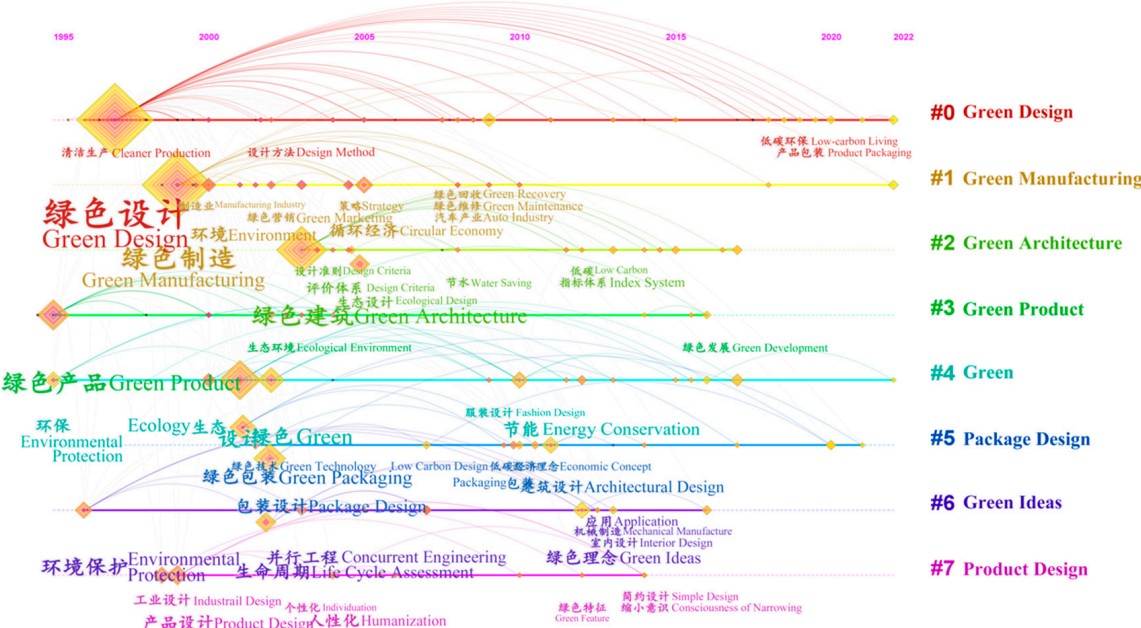

**Figure 9.** Knowledge Graph of Green Design in CNKI database.

First, the high-frequency subset in the green design group consists of terms such as sustainable development, recycling, hierarchical bonding, and graphic design. Second, the green manufacturing group includes product life cycle, green maintenance, the automotive industry, and topologic theory. Third, the green subset covers green building, green manufacturing, implementation strategy, northern region, and renewable energy.

The hotspots of green design research in China are long-term and stable, involving industrial, design, social, and architectural fields. Furthermore, it indicates that green design in China mainly revolves around livelihood industries, and this phenomenon is more related to the importance China attaches to sustainable development. However, it is shown from the comparison of the WoS database that although green design in China is covered in several disciplines, such as chemistry or engineering, there needs to be more collaborative research across disciplines.

## 9. Conclusions

In this study, we employ the CiteSpace econometrics algorithm to analyze green design literature from CSSCI core papers in the WoS and CNKI databases. This approach overcomes the limitations of relying on a single database for a specific research discipline, yielding more diverse and comprehensive results. We summarize key elements, development hotspots, and knowledge structure in the fifty-year development history of green design within a global context. China has the highest volume of published research worldwide. By analyzing Chinese literature, we enhance the comprehensiveness of the green design database, providing practical implications for green design in China and globally. Our findings reveal the following:

1. The research heat and volume of green design are increasing annually, demonstrating its growing importance as a tool for sustainable development. Green design experienced rapid growth around 2015, with high collaboration rates, citations, and publication volume in various sub-fields, such as energy conservation and material selection.

2. WoS hotspots focus on solving practical problems in sub-disciplines, primarily chemical, experimental, and case-based. Chinese hotspots emphasize design disciplines, mainly concentrating on design processes, management, and theory.

3. The WoS database centers on natural and social science categories, while the Chinese database highlights art and design theory research.

The clustering, knowledge, and keyword graphs from the WoS database indicate good international academic communication in green design, with four research chains formed by various universities and research institutions. Green design exhibits a solid academic foundation, allowing future scholars to develop diverse research frameworks. Practical research on pollution reduction methods, such as energy conservation, technological innovation, environmental protection, and management, dominate research keywords. However, more research on green design's theoretical concepts is needed, as no basic scientific research on principles and laws with green design as the core has been formed.

The clustering, knowledge, and keyword graphs from CNKI's CSSCI literature show that green design in China started late but formed hot topics earlier than the WoS database. This may be due to Chinese environmental protection initiatives and the rise of design disciplines. Hot topic areas differ, mainly involving design, architecture, and management. Connectivity between green design research within CNKI and WoS is limited, but Chinese scholars have made significant academic contributions to WoS. This suggests CNKI's literature focuses on art and design research, while the WoS database emphasizes natural science research.

The overall research trend of green design is increasing obviously, and it is easy to see that it has been paid attention to by many international scholars. The hot spots in the discipline form specific patterns as follows: (i) Green management aspect. Through the rational planning program, the green cycle and improved efficiency and saving energy are practiced to achieve the goal of green design. (ii) Green building and nature design. Green building materials, advanced technology, and the right balance of light, weather, and vegetation are wielded to build a human living environment, thus achieving the green goal of reducing pollution and coexisting with nature. (iii) Sustainable materials and waste recycling technology. Scholars have found renewable green materials to replace traditional materials and reuse waste products to achieve the green goal of reducing and zero waste. (iv) Green economy. In order to achieve the goal of cost reduction and pollution, product packaging has to be recycled and transportation reduced.

The overall research trend of green design is increasing significantly, attracting attention from international scholars. The discipline's hotspots form specific patterns, including green management, green building and nature design, sustainable materials and waste recycling technology, and green economy. Green design is an interdisciplinary area of expertise crucial to sustainable development. However, as an independent research topic, green design lacks basic research, including discipline definition, philosophical discussion, and research revealing its core nature.

This may prevent the green design from shaping future healthy human and environmental development prospects. However, through the study's research, the green design future ensures the feasibility of a wide range of designs. Based on the findings of this paper, green design will challenge optimizing existing production conditions, economic conditions, resource management, and scientific research and development.

According to the previous findings, the future research focus of green design tends to be (1) Optimization of production and people management. The development of new management guidelines and production design to ensure sustainable green development in terms of time, materials, and human resources. (2) Human health and environmental well-being. Green design will be based on green design and incorporate elements such as architecture, environmental planning, and materials into the design principles and process. (3) Development of green technologies and tools. We seek new materials to replace traditional processes and consumables to ensure sustainable development. (4) Society and culture. Promote and nurture the concept of a green design through different cultural contexts. To ensure that the research field of green design has a positive development in interdisciplinary learning.

This study has limitations, as some academic conferences involve oral presentations or online conference discussions, making their content inaccessible in web-based databases. Future research should build upon the theoretical foundation established in this study and

explore relevant academic conferences and specialized books to conduct more extensive research. This will help address the obstacles and challenges encountered in the development of green design. Finally, the authors call on more designers, educators, artists, and philosophers to join the study of green design, providing more creativity, fundamental disciplinary research, and theoretical exploration.

**Author Contributions:** Conceptualization, Y.D.; Methodology, Y.D.; Data curation, Y.D.; Writing—original draft, Y.D.; Writing—review & editing, H.M.; Visualization, H.M. All authors have read and agreed to the published version of the manuscript.

**Funding:** China Postdoctoral Science Foundation: 043200000; China Education Association for International Exchange: 20225440015; Scientific research start-up funds of Guangdong Ocean University: 060302162101.

**Institutional Review Board Statement:** Not applicable.

**Informed Consent Statement:** Informed consent was obtained from all subjects involved in the study.

**Data Availability Statement:** No new data were created.

**Conflicts of Interest:** The authors declare no conflict of interest.

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
