# Peer review of "A Study on the Visual Communication and Development of Green Design: A Cross-Database Prediction Analysis from 1972 to 2022"

_sustainability, doi:10.3390/su15086359_

Round 1

Reviewer 1 Report (Previous Reviewer 1)

I included your paper with comments directly in the manuscript.

The English language used is much improved and now the manuscript is readable. A few minor issues remain where I marked most of those obvious ones. In one place a full paragraph was more or less repeated in need of adjustment (p.8. row.270-280).  

The paper would be improved by linking this paper's motivation with previous research/scholars (see comments in file). It would be a better read to introduce some of the main trends for green design in the introduction, placing the significance of your manuscripts contribution in context and making the motivation sanctioned.

Good compiled literature review 

Figure 2 may be including some errors: the scale in relation to the green line does not make sense. The green line has higher values where I expect lower values. The green line is above the orange line even if the text says the amount of research represented by the orange line should be more than for the green.  Please check this up.

Improvements could be done to the analysis presented in section 8.1 to be more clear and to the point.

Author Response

Reviewer 2 Report (Previous Reviewer 3)

The author has revised the paper according to my comments. 

Author Response

Thank you for your valuable feedback on my manuscript. I truly appreciate your time and effort in reviewing my paper.

This manuscript is a resubmission of an earlier submission. The following is a list of the peer review reports and author responses from that submission.

Round 1

Reviewer 1 Report

The paper suffers from several weaknesses as per the current version. Poor English throughout the manuscript makes several sentences incomplete and without meaning, or without meaning related to the message of the paper. There are also multiple typo errors in need of correction. This makes the manuscript inappropriate for communicating the intended content. The manuscript must be proofread by both an expert in English and the research it considers to fix this. It is essential to improve the language to be even considered for publication.

Moreover, the introduction lacks support from relevant research, using seven citations in total, and some of them are thirty years old to provide for motivating the proposed study, not ok.

The study design, which uses citespace software to analyze publications, it identifies papers from some selected indexes at the Web of Science and from the database of China National Knowledge Infrastructure (CNKI). This could be an interesting approach. However, conclusions summarize a Country ranking of total co-citations, where the number of publications orders the top three countries: China, USA, and India. This makes me wonder: How much is the selection of the databases affecting the resulting conclusions? 

The quality of the figures needs to be improved. The clarity of language needs to improve to interpret the content of the paper. In a future version of your paper, I strongly suggest using complete clear sentences that are carrying the message through, better with fewer words well presented than many that instead confuse the reader.  This will help review the quality of the presented research, currently not possible.

Reviewer 2 Report

Title is very promising! Abstract does not represent what is said later in the article. In section 1 clear definition of ecodesign and green missing, so no clear reference for 'space to improve' (line 42).  Interesting statement on line 52/53 : for many readers (not for me clarification of this needed. Line 76 - 84  inresting (and true), but what does this mean for Eco/green design ? Section 2 no comment. Section 3 explain what WoS is (a data bank?) Very important question : where is Europe is this statistics? (Europe has been the leader in Eco/green design at least in the period 1992-2006, miss this also in the reference list. Section 4 Fig 3 unclear, on what conclusion that USA is a leader (line p 194) is based on What is the conclusion of section 4. Section 5.1 several key words have as such nothing to do with eco/green design (methylene bluegeberic algorithm, ion, decision) . What do colors in the figure represent. 5.2. 'silhouette'drops out of the air, what does it mean? again words topic which have nothing to do with ecodesign like malachite green, protein engineering, enhanced analitical quality.  Discussion of all this is a mess! (As a result?)Again no clear conclusion of this section. Section 6 is more clear however, explain much better what these cluster diagrams are supposed to mean. Section 7 core (fig 8) is unreadable; does not support text. . Section 8 no comment. Section 9 ; the conclusion drop out of the air, not supported /substantiated by evidence. 

Section 2 no comment. 

Reviewer 3 Report

1. This manuscript needs more work on how the sentences are put together because there are a few grammar mistakes.
2. The motivation and contribution of the proposed work are not given in the manuscript. Authors should include a subsection specifying the proposed work's motivation and objective. The novelty of the proposed work should be explicitly highlighted. It isn't easy to see the Novelty of this work.
3. Related work and existing literature need to be better presented. It is suitable to describe existing studies, focus areas and problems identified from existing literature.
4. State the weakness of the developed work.
5. References must follow the style. Especially, the style of citing references needs careful editing and must be consistent.

6. It is suggested that the passive voice should be used in writing.
